# Healthcare Professional Students’ Perspectives on Substance Use Disorders and Stigma: A Qualitative Study

**DOI:** 10.3390/ijerph19052776

**Published:** 2022-02-27

**Authors:** Alina Cernasev, Kiki M. Kline, Rachel Elizabeth Barenie, Kenneth C. Hohmeier, Steven Stewart, Shandra S. Forrest-Bank

**Affiliations:** 1College of Pharmacy, University of Tennessee Health Science Center, 301 S. Perimeter Park Drive, Suite 220, Nashville, TN 37211, USA; khohmeie@uthsc.edu (K.C.H.); sstewa46@uthsc.edu (S.S.); 2College of Social Work, University of Tennessee, Knoxville, TN 37996, USA; kkline5@vols.utk.edu; 3College of Pharmacy, University of Tennessee Health Science Center, Memphis, TN 38163, USA; rbarenie@uthsc.edu; 4Director of the Social Work Office of Research and Public Service (SWORPS), College of Social Work, University of Tennessee, Knoxville, TN 37996, USA; sforres6@utk.edu

**Keywords:** substance use disorder, stigma, medical students, pharmacy students

## Abstract

Background: Access to and quality of care for Substance Use Disorders (SUDs) remain a major public health issue. Stigma associated with SUDs contributes to the gap between the number of patients who need treatment and the much smaller fraction that receive it. Healthcare professional students are future care providers; an opportunity exists to characterize their collective perspectives on patients with SUDs and how that informs the care they provide. Methods: Healthcare professional students participated in online, semi-structured focus group (FGs) between March and April 2021. The FGs were conducted until thematic saturation was achieved. All verbatim transcripts were analyzed applying Thematic Analysis using Dedoose^®^ qualitative software. Inductive codes were grouped into categories based on similarities that facilitated the emergence of themes. Results: Thematic Analysis revealed one theme (1) Decreasing stigma among healthcare professionals by viewing substance use disorder as a disease; and two sub-themes: Subtheme 1a: Relating with the patients, “It could be me…”; Subtheme 1b: Interactions with patients, “We just don’t know exactly how to counsel these patients…” These themes describe how future healthcare professionals might perceive and approach patients with SUDs and highlight the importance of SUD training in the curriculum. Conclusion: Medical and pharmacy students are uniquely positioned to apply critical thinking from their didactic training to their real-world clinical experiences, and their collective perspectives inform gaps in training and opportunities to develop best practices for SUD care. An opportunity exists to leverage these findings in order to train future healthcare professionals to ensure access to and quality of SUD care.

## 1. Introduction

Access to and quality of care for Substance Use Disorders (SUDs) remains a major public health issue. A recent report by the Substance Abuse and Mental Health Services Administration (SAMHSA) indicates that approximately 14.9% of persons 12 years or older needed treatment in the last year, but only 1.4% actually received any treatment [1]. This highlights the large gap that exists between patients who need treatment and those who actually receive it. Patients who misuse substances, such as opioids, are at risk for drug-related overdoses and even death. The Centers for Disease Control and Prevention’s (CDC) report stated that there were 70,669 drug overdose deaths in October 2019, with a large majority of these overdoses attributable to opioids [2]. The coronavirus pandemic has exacerbated drug overdose deaths in the United States, with an estimated 30% increase from October 2019 to October 2020 in opioid-involved overdose deaths [2]. This situation is more alarming in Tennessee where this increase is expected to be as high as 42.5%, or 12.5% above the expected national average [2].

The social and economic diversity within Tennessee communities presents unique challenges related to the opioid epidemic and public health practice. Tennessee represents a state with diverse geography, spanning from Memphis (large minority population) to Nashville (urban and diverse), to Knoxville (a much smaller city), to numerous rural Appalachian areas. Unfortunately, Tennessee continues to be one of the worst states impacted by the opioid epidemic, with the number of heroin overdose-related deaths increasing by over 300% in the last three years [3].

Safe and effective treatment exists for several types of SUDs. For example, clinical practice guidelines for the management of Opioid Use Disorder (OUD), which is a type of SUD, recommend medication in combination with psychosocial support. Evidence of the effectiveness of SUD treatments shows that medications, such as buprenorphine and methadone, were found to significantly decrease return to use rates compared to behavioral therapy alone [4]. In addition, patients who were maintained on medication treatment for OUD for longer than one week had a 28% reduction in their risk for opioid-related death [5]. This highlights the importance of health care practitioners being informed about the most effective interventions for people with SUDs, including medication for OUD.

### 1.1. Preparation of Pharmacy and Medical Students to Treat Patients with SUDs

The expertise of multiple providers is often necessary to provide the full scope of care a patient with a SUD needs. For example, it is not unlikely that a patient will require treatment for diabetes from a physician, along with a social worker who leverages support and resources, a counselor who treats substance misuse, a psychotherapist for co-occurring mental health disorders, and a pharmacist who provides medication. All healthcare providers are likely to treat patients who have active substance misuse and therefore must be trained to recognize symptoms and interact effectively with patients in order to adequately assess treatment needs and ensure that patients access appropriate care. Otherwise, patients are likely to receive suboptimal care and become more ill. For example, a retrospective study of patients hospitalized with infective endocarditis associated with injection drug use found high rates of readmission, recurrent disease, and death [6]. Less than a quarter (23.7%) of the patients had received addiction consultation, addiction was mentioned in only 55.9% of the discharge summary plans, and 7.8% of the patients had a plan to receive medication-assisted treatment [6]. Research has also demonstrated that physician–pharmacist collaborations are associated with improved treatment outcomes for patients with OUD [7].

The focus in recent years has shifted to include training on SUDs in the curriculum, due to their prevalence. However, it is not clear to what extent, when in the curriculum, and how this training is being assessed. The Doctor of Pharmacy (Pharm.D.) and Doctor of Medicine curriculum at the University of Tennessee Health Science Center is comprised of a four year curriculum, including both didactic classes and experiential education. During the Pharm.D. curriculum, the students can enroll in elective courses that explore various disease states or provide additional interprofessional activities. Student pharmacists study pharmacotherapy, medicinal chemistry, and pharmacology of opioids throughout the Pharm.D. curriculum. Education in practices to engage effectively with patients is limited, and if received is usually incorporated into the therapeutic modules throughout the curriculum [8]. It is not clear how well the training Pharm D. and M.D. students receive prepares them for working with patients with SUDs.

### 1.2. Stigma toward Patients with SUDs

One of the most influential factors that limits patients from seeking and accessing the help that they need is stigma. Stigma can be understood as having three components: stereotypes, prejudice, and discrimination. A stereotype is a generalized, commonly held belief about a group of people that represents an oversimplified and automatically accepted opinion and a prejudiced attitude. Prejudice is a hostile and irrational attitude directed toward a person based on the perception that they represent a certain group, and on preconceived notions based on that group affiliation. Discrimination occurs when unjustified distinctions are made between people, due to the group to which they belong or are perceived to belong to [9]. Stigma occurs both as publicly held and externally imposed and as internalized stigma [9].

Common themes of stigmatization toward people with SUDs manifest in health care settings and systems. For example, there is a misconception that continued use of substances despite severe consequences is a willful choice rather than a symptom of physiological changes that have occurred in the brain. [10] In addition, treatment for OUD tends to occur separately from other healthcare entities as though it is not a medical problem. Extremely common also is the use of negatively-connotated language when referring to patients with OUD [11]. Particularly strong stigma exists toward medication assisted treatment. A systematic review of 972 studies on stigma regarding interventions involving medications for OUD found that lack of training and preference for abstinence rather than medication influenced the stigma held by healthcare providers [12].

There is a growing recognition that stigma toward patients with SUDs is a problem that interferes with health care and must be addressed. One recent study conducted with physicians, nurses, and medical students surveyed their views on SUD and found that medical students more frequently reported that “more work needs to occur to minimize stigma related to substance use disorder” when compared to nurses [13]. These perceptions are echoed by other research that indicates stigma towards people with SUDs exist within the medical profession. A study by Fischbein and Bonfine (2019) examined the differences in attitudes between pharmacy and medical students on general mental health concerns and found greater levels of stigmatized perceptions by pharmacy students when asked for their level of agreement on the statement “Most people think less of a person who has received mental health treatment.” [14]. Another recent study that assessed third-year student pharmacists during a simulated opioid counseling session found that many students lacked confidence in their verbal and nonverbal communication skills when working with OUD patients [15]. These students further reported the need for additional education and resources on how to best communicate with patients about related sensitive topics [15]. An explanation for this could the lack of relatable social engagement practices physicians engage in with their patients, which has been demonstrated to be an effective strategy to reduce stigma related to mental illness [16].

### 1.3. The Current Study

Given the gap in accessing treatment and the role of stigma in perpetuating suboptimal care for patients with SUDs, there is a critical need for investigating health care students’ experiences of treating patients with SUDs, and preparing them to provide services that transcend stigma. The collaboration between prescribing practitioner and pharmacist may be a key to identifying barriers and implementing initiatives that improve access to and quality of care provided, especially during times of global pandemic, which has exacerbated an already existing epidemic. However, data are lacking about the collective voice of medical and pharmacy students and their perspectives on and confidence about managing patients with SUD. Thus, the objective of this study was to characterize future healthcare practitioner perspectives about what works best to reduce stigma and to interact effectively with patients with SUDs when providing care.

## 2. Methods

### 2.1. Study Design, Participants and Data Collection

We employed qualitative focus group (FG) methodology to understand and characterize healthcare students’ perception of SUDs and the influence of stigma on care provided, in order to inform strategies to improve access to and quality care for patients with SUDs. Participants in this study were recruited via email from the University of Tennessee Health Science Center (UTHSC). The College of Pharmacy has three campuses: Memphis, Nashville, and Knoxville; the College of Medicine has one campus in Memphis.

Participants self-selected their participation and inclusion criteria were (1) adults, (2) enrolled in the professional program at the time of study, (3) English speakers, (4) familiarity with the topic, and (5) willingness to discuss their opinion. Participants received a $30 gift card as compensation for their participation. The University of Tennessee Institutional Review Board approved this study (IRB: 21-07977-XM; 1 March 2021).

The FG guide was developed based on Stigma Conceptualization [9] to examine participant perceptions and experiences regarding stigma when caring for patients. The FGs comprised of open-ended questions, probing views on stigma associated with SUDs. All FGs were conducted by two researchers (AC, SFB).

Verbal informed consent was obtained prior to the FGs, in which the procedures were explained to participants and any questions about participation were answered. All the participants agreed to be audio recorded and verbatim transcription was conducted by an objective professional company to minimize bias. The data were collected from March to April 2021 when thematic saturation was achieved [17]. In contrast to quantitative methods in which power or a statistical formula is used to determine the sample size, qualitative research uses “theoretical saturation” or “redundancy” [17]. To achieve theoretical saturation, the team continued recruitment of the participants and conducted FGs until no new insights on the topic were presented in the FGs [17].

All FGs took place virtually to accommodate circumstances stemming from the COVID-19 pandemic [18]. The FGs started with the following background information collected from participants: age, college attended, year in the college, race/ethnicity, and gender. Duration of the FGs ranged from one to two hours. The first question of the FG was, “How would you define stigma in your own words?” This opening question allowed the participants to bring the topic of stigma and their experiences in health care settings to the forefront, and get the conversation flowing. Another central instruction was “Describe a situation when you interacted with a patient who was diagnosed with substance use disorder.” The FGs were semi-structured so the discussions among participants could emerge naturally, while covering the intended scope at the same time. Participants were encouraged to talk about what their observations were regarding what worked best to reduce stigma and help patients with SUDs. The data in the transcriptions that were coded as being solution-oriented (rather than describing the problem) were analyzed in the current study.

During the transcription process, a supervisor verified the preliminary audio recordings against the transcript for accuracy. A research team member also listened to two audio recordings in order to ensure the accuracy of the data (AC). This method served as a form of verification before data analysis was initiated.

### 2.2. Data Analysis and Establishing Trustworthiness

Themes and subthemes were identified using thematic analysis with an inductive approach [19]. A reflexive process was followed for conducting a thorough and transparent data analysis [19]. After familiarization with the data, the transcripts were coded with verbatim pieces of text and then preliminary codes were generated. Similar codes were grouped into categories [19]. All the categories were clustered and analyzed to uncover the major themes. Two researchers performed a line-by-line reading of the first two FG transcripts, after familiarizing themselves with the corpus of data (SS, AC). After review, the researchers collaborated to refine and define the initial codes as well as to identify emergent codes.

A process of independent review and open discussion was implemented to clarify and revise the coding frame until consensus was reached [20]. The coding process was facilitated by qualitative software, Dedoose^®^ (v2.0, Manhattan Beach, CA, USA), which was used for generating initial codes and developing and reviewing themes. Another two researchers reviewed the final codes and one researcher identified common and recurrent themes and subthemes (KK, SFB) [20].

The researchers used Yardley’s criteria to ensure that the quality and rigor of qualitative research were met [21]. For commitment and rigor criteria, all stages outlined by Braun and Clarke were followed [19]. Furthermore, the researchers followed the consolidated criteria for reporting qualitative research (COREQ) throughout the study design and data analysis [22]. The credibility criterion was achieved by having three researchers to check the identified codes and the emergent themes (AC, KK, SFB). Lastly, reflective memos were written by the researchers throughout the process of data collection and analysis [23]. All authors involved in the data collection and analysis have disclosed their positions regarding the research topic through a memo [23]. A synthesis of the data collection and analysis can be found in Figure 1.

## 3. Results

A total of five FGs were conducted between March and April 2021, with a total of 31 participants. Of all 31 participants, the mean age was 26.7 years old, 10 identified as male, and 21 identified as female. The majority of the participants (n = 17) were from the College of Pharmacy (COP) and14 were from the College of Medicine (COM). All the COM participants were in the fourth year of their degree, while the COP participants represented second, third, and fourth years.

This manuscript focuses on one emergent theme and two subthemes aligning with the study objectives. The theme and sub-themes will be described in conjunction with participant quotations and excerpts from the FGs.


*Theme 1: Decreasing stigma in healthcare professionals by viewing substance use disorder as a disease*


Th9s overall theme explains how the healthcare professional participants recommended decreasing stigma when working with their SUD patients by understanding them as suffering from a disease. They tended to convey a perspective that, by relating personally to the circumstances that patients with SUDs experience, they were able to reconceptualize stereotypes as human struggles. In doing so, they were able intentionally not to allow themselves to automatically ascribe negative judgement and attitudes that otherwise may have acted as a barrier to interactions with the patients. Two subthemes further explain healthcare professional participants’ experiences with this reconceptualization of SUD patients.

The two subthemes that emerged from data analysis were (1a) relating to the patients, “It could be me” and (1b) without personal judgement, “They are just people with a physiological dependence on something.” These subthemes further expand on the participant’s narrative regarding how patients should be approached, i.e., as a patient with a disease.


*Subtheme 1a: Relating to the patients, “It could be me…”*


The first sub-theme underlines how the participants considered both the patient and practitioner as persons who could experience SUD as a disease. The participants expressed their understanding of the reality that developing a SUD, even as a practitioner, was possible. This understanding fostered participants’ belief that their patient may be able to see themselves in the practitioner, thus establishing a closer patient–practitioner relationship. For example, a participant summarized this interaction by stating:

“… and the degree of separation between myself and (them), it’s very small. You know, it could be me. So, I think that we need to learn, especially learn that it could very well be us …”(FG #2, Participant #18, Male, COM)

Other participants elaborated on the prevalence of medical practitioners’ experience of SUDs, especially pharmacists. Specifically, one participant spoke directly about this issue, expressing that the reason why this is so frequent was because the pharmacists are the “*gatekeepers of the drugs*” and that they have “*ready access to (the drugs).*” This participant expounded by saying that “there is that temptation” because of the nature of the job. As she reflected:

“…it’s really prevalent among healthcare professionals, substance abuse disorders, I think I want to say like almost upwards of 30% of individuals. And so, we’re not immune to it. And especially pharmacist, I mean, we’re the gatekeepers of the drugs, we have ready access to them, and so, if there is that temptation, it’s very easy to succumb to it. And so, yeah, we’re no different from anyone else.”(FG #2, Participant #6, Female, COP)

Further, many of the medical students spoke about the idea that a person with a SUD or who is misusing substances may not always be readily apparent, and if it is a practitioner with an SUD or misusing, that person may be overlooked due to the nature of their profession. A female participant echoed many of the other students by providing an example of how a practitioner with a SUD may present:

“One of our symposiums (had a speaker who) was a pharmacist, who went through the rehabilitation process and getting his license back to practice … He was the pharmacist that showed up early, stayed late, (and) did these things to get what he wanted from the pharmacy, but also appeared to be like a really hardworking pharmacist. He looked successful on the outside, like a really hardworking guy, but then he had this underlying problem.”(FG #3, Participant #10, Female, COM)

However, even though the participants recognized the prevalence of SUDs, they reported that they did not know enough about the disease. This impacted how comfortable they felt working with patients with SUDs, and participants reported preferring more on-the-job training. They also mentioned wanting to learn more about their own risk for developing an SUD. Participant #17 expressed her feelings about this by saying “…that’s why I’m so uncomfortable, we don’t feel like we know enough about (it).” She also summarized the participants’ consensus on this topic by providing recommendations for training:

“Not necessarily understanding what they’re taking and why their taking it in terms of misuse of drugs, but (we need to learn) more how we can relate to them.”(FG #1, Participant #17, Female, COP)

Participants emphasized that having the ability to relate to patients and meet patients where they are may be an opportunity to offer greater support and empathy. The participants indicated that this approach would allow the patients to see themselves in the practitioner, which may build a stronger therapeutic relationship and advance help-seeking behavior for their SUD. Participant #2 gave an example of how to do this in the following excerpt:

“So just small things like checking up on them … When I ask (them) how (they’re) doing, and (they) tell me, ‘I’m having this issue with my kid, he’s been acting out,’ when I see (them) again, I’m asking ‘So how is your son, is he still acting out?’ … It just helps to show people that someone cares, (that) I can be a source of comfort for someone in the smallest way that I can be.”(FG #2, Participant #2, Female, COP)

Sub-theme 1b: Without personal judgement: “They are just people with a physiological dependence on something…”

The second sub-theme captures the participants’ beliefs that patients should be seen as individuals with a physiological dependence and not be judged for their SUD. The participants used a variety of other terms that they believed to have the same meaning such as “illness,” “disease,” “sickness,” a substance “their body needs.” The data in this subtheme recognized the social stigmas that are associated with their patient’s experiences, and that as healthcare professionals they need to see beyond. A SUD is similar to any other disease; with appropriate and pragmatic interventions, it can be treated. Participant #19 highlights the core of this theme with the following quote:

“I think the biggest thing is remembering that these (individuals have) a physiological dependency. Their body needs (substances), especially with opioids and alcohol withdrawal. (This) is one of the big first steps in approaching it like a disease and something that you should be able to treat.”(FG #5, Participant #19, Male, FG #5, COM)

However, even though people in the healthcare profession seem to recognize SUDs as a disease, there is still stigma present in medical settings. Practitioners are not immune to societal perceptions that a SUD is a ‘moral weakness’ and is thus a personal choice. Participant #11 explained, *“I think a lot of times, as healthcare professionals, we forget to look at these people as patients who are just dealing with an illness.” (FG #1, Participant #11, Female, COP)*

Participants expressed the importance of recognizing that stigma exists, treating their patients with respect, and understanding the severity of their symptoms. Participant #12 shared that, because of the risks that come with SUDs, practitioners have a responsibility to understand the diagnostic criteria and manifesting symptoms. Participant #12 states:

“Part of our job is to set the tone and to say, ‘I’m going to treat you the same regardless of how you treat me, and I’m not going to be cavalier with your complaints ...”(FG #5, Participant #12, Male, COM)

The participants in FG #1 shared the idea that a practitioner must approach patient care through joint decision-making between the patient and practitioner in order to best meet the patients’ needs. FG #1 discussed that, because of the nature of SUDs as a serious illness, there is little room for personal opinions on what is right or wrong. Participant #3 noted that a harm reduction approach may be helpful in connecting patients to treatment. She provided the example of her experience at a needle exchange program during her medical training:

“We have everyone coming in with substance use disorders to get clean needles and the volunteers there would ask every single person every time they (came in), ‘Are you ready to get help? We have resources for you.”(FG #1, Participant #3, Female, COM)

She further discussed how the volunteers interacted with the individuals who came in for the needle exchange, noting their empathetic and understanding demeanor. She believed that their approach led to positive outcomes for patients misusing drugs:

“(The medical volunteers) were already treating them with respect in saying, ‘I know what you’re going through and I’m not judging you for it, but I have resources to help when you’re ready.’ They have gotten several people into recovery programs (because of this) and a lot of them have been very successful.”(FG #1; Participant #3, Female, COM)

Moreover, when the practitioners view SUDs as an illness outside of the patient’s control, it can help relieve some of the internalized stigma that patients may have about their problem. Participants discussed how practitioners can create a counseling-type relationship with their patients, informing them about the nature of their substance misuse and establishing trust. For example, one of the participants described how patients can have negative opinions about interacting with practitioners and treatment. Participants provided feedback about how medical practitioners can interact with their patients with SUDs in order to reduce stigma:

“Explain to them that it’s a sickness they’re dealing with. Don’t make them feel bad for the choices they’ve made because 9 times out of 10 they already do (that to themselves) and that’s what makes them not go get help because they don’t want to be judged … really try to get across that (substance misuse) is a sickness, just like if you had hypertension or high cholesterol, it’s the same type of thing.”(FG #1, Participant #11, Female, COM)


*Subtheme 1b: Interactions with patients, “We just don’t know exactly how to counsel these patients…”*


This theme describes the participants’ perceptions on interacting with patients with a SUD. For example, during the FG discussion, one participant mentioned the importance of counseling, but stated that this can sometimes be challenging for various reasons, which is reflected in the statement below. It also became clear that, although the training provided in the curriculum is extensive, more interactions with patients, listening to their stories for cues about how they want to be approached, are needed.

“…We just don’t know exactly how to counsel these patients. Because a lot of them we see, at least in a retail setting, I see them every seven days, and I think that it would be really helpful to maybe have someone that has a substance use disorder come in and say, hey, this is how I would like to be monitored every week, these are the conversations that I want to have, this is how you can check in and see how I’m doing. Because we do have to counsel them every seven days and it’s kind of like, well, I don’t always know exactly to cover and what to talk about.”(FG #1, Participant #3, Female, COP)

The following quotation emphasized the value of learning from a patient with a SUD. The participant commented:

“…I think the school has trained us to, you know, at least in (Course Name)… to speak professionally and be empathetic. They like to teach empathy and compassion, so they’re always like, be empathetic with the patients, compassionate...”(FG #2, Participant #22, Male, COP)

Some participants discussed the importance of transparency from the practitioner perspective. For example, some patients may feel powerless when the healthcare team reveals information about their substance misuse from laboratory findings. Some patients may feel pressured to keep their use a secret for fear of criminal punishments. As one participant explains:

“…So we (medical team) always ask them if they use cocaine. And then, sometimes they say yes, sometimes the patient says no. And the doctor always orders the urine drug screen. And the patient said no, and the urine drug screen was positive. And when pressed about that, some of my patients have said, oh, yeah, I did use cocaine, and some of them actually said, oh, my brother uses cocaine, I think it may have gotten on my shirt or something like that. And so, I feel like there’s a sense of shame and a sense of hesitancy from the patients. And I find that, sometimes, the patient will always deny it, but a lot of the times the patient will admit it when the attending and the team presents it in like a nice manner and not threatening and accusatory, which has worked out a lot actually.“(FG #3, Participant #1, Male, COM)

Several participants also highlighted the importance of listening to the patients’ needs and establishing engagement with them so they will feel comfortable sharing their life stories with the healthcare team. One participant indicated feeling closer to a patient because she was there in the room and focused her attention on the patient’s story, which allowed the patient to become the center of the care provided. One participant elaborated:

“Listening is incredibly important. If you tell people what they should do, that’s not going to get you very far. So just I listen to these patients. I try to understand what’s going on in their life, what led them to this point. What’s your goal? What do you want out of this experience?”(FG #2, Participant #6, Female, COP)

Participants shared the idea that verbal interactions between the practitioner and the patient should be respectful under all circumstances. Practitioners should demonstrate dignity in dealing with patients and should communicate that they are willing and able to understand their situation. Participants also mentioned that, in their experience, fostering direct lines of communication led to better outcomes for the patient. A participant revealed the significance of this type of interaction between practitioner and patient in a crisis situation:

“We had a patient flown in from rural Tennessee … he had swallowed several bags of I believe methamphetamines, so he was being flown … for emergency care. I remember in preparation for him coming, he was an older teenager, (in general) pediatricians are a little scared of teenagers because they’re older, and when it comes to substance use, it’s just not something you see very often in a pediatric setting… When he flew in he was thankfully stable... The next night, however, he had an episode of very severe agitation … and he really only calmed down when (we) used a very calm tone.”(FG #2, Participant #15, Female, COM)

## 4. Discussion

Among 31 FG participants, two major themes were identified: (1) Healthcare professional students’ perceptions of Substance Use Disorders, and (2) Interactions with patients, “We just don’t know exactly how to counsel these patients…” These themes describe how future healthcare professionals might perceive and approach patients with SUDs and highlight the importance of SUD training in the curriculum. An opportunity exists to leverage these findings in order to train future healthcare professionals to eliminate stigma and to improve SUD care.

Prior research has identified stigma as a major barrier to SUD care and has demonstrated that healthcare practitioners may provide lower quality care to patients with SUDs due to that stigma. However, whether that might also be observed among future physicians and pharmacists had not been explored. Participants in this study discussed their perspectives on managing SUD patients, and medical and pharmacy students tend to be aware of the presence of stigma and the barrier it can present in being able to optimally care for patients. Participants who identified and empathized with patients with a SUD seemed not only able to engage more effectively with patients, but also be more confident in their capacity to help. A novel finding of this study is the collective perspective of participants and that their own potential stigma was mitigated by personally relating to the circumstances of patients with a SUD.

While earlier studies were conducted with medical students or, separately with pharmacy students, these prior findings did not focus on the interprofessional perspective [24,25]. For example, a survey of interprofessional students (pharmacy and dentistry) highlighted the significance of understanding OUD and patient care [26]. Several participants highlighted that both the patient and practitioner could experience a SUD [26]. Our findings revealed that the participants identified the importance of instruction and experiential learning opportunities directed towards how to effectively treat patients who present with a SUD.

Our findings also show that, for some participants, there was a challenge regarding counseling patients with SUD. Previous studies suggest that student pharmacists lacked confidence in counseling patients about opioid-specific risks such as dependence, addiction, or overdose [15]. Our findings not only echo the prior literature that a need exists to improve student confidence in communication with patients with SUDs, but provide examples of specific skills that should be focused on. For example, participants in our study highlighted the need for additional training in listening and empathizing. These skills could be added to assessment rubrics to reinforce their importance in patient interaction.

### Strengths and Limitations

To our knowledge, this is the first study that presents the collective perspectives of SUD and stigma from a cohort comprised of medical and pharmacy students. Unfortunately, although this study invited other healthcare professional students to participate, some did not respond to our recruitment strategy in time to facilitate participation. Therefore, future studies may consider including focus group participants containing a more diverse sample of healthcare students.

Although the participants were from a homogenous geographic location, ethnicity, and gender identity, this sample enabled this study to examine the participants’ perceptions of the topic in some depth. It would be valuable to conduct further work inclusive of participants from varying demographic backgrounds and geographic areas of the US.

## 5. Conclusions

Students in professional healthcare programs provide an important perspective for informing gaps and best practices for care, since they are uniquely positioned to apply critical thinking from their didactic training to their real-world clinical experiences. This study reveals that students felt that stigma toward people with SUDs is a barrier to accessing treatment and receiving adequate care. Students felt that healthcare professionals who do not understand the nature of SUDs, and do not intentionally combat their own stigma, are likely to provide sub-optimal care and perpetuate stigma. Intentional instruction about SUD and engagement skills that facilitate genuine empathy and nonjudgmental attitudes may reduce barriers and allow patients to be more honest about their experiences.

## Figures and Tables

**Figure 1 ijerph-19-02776-f001:**
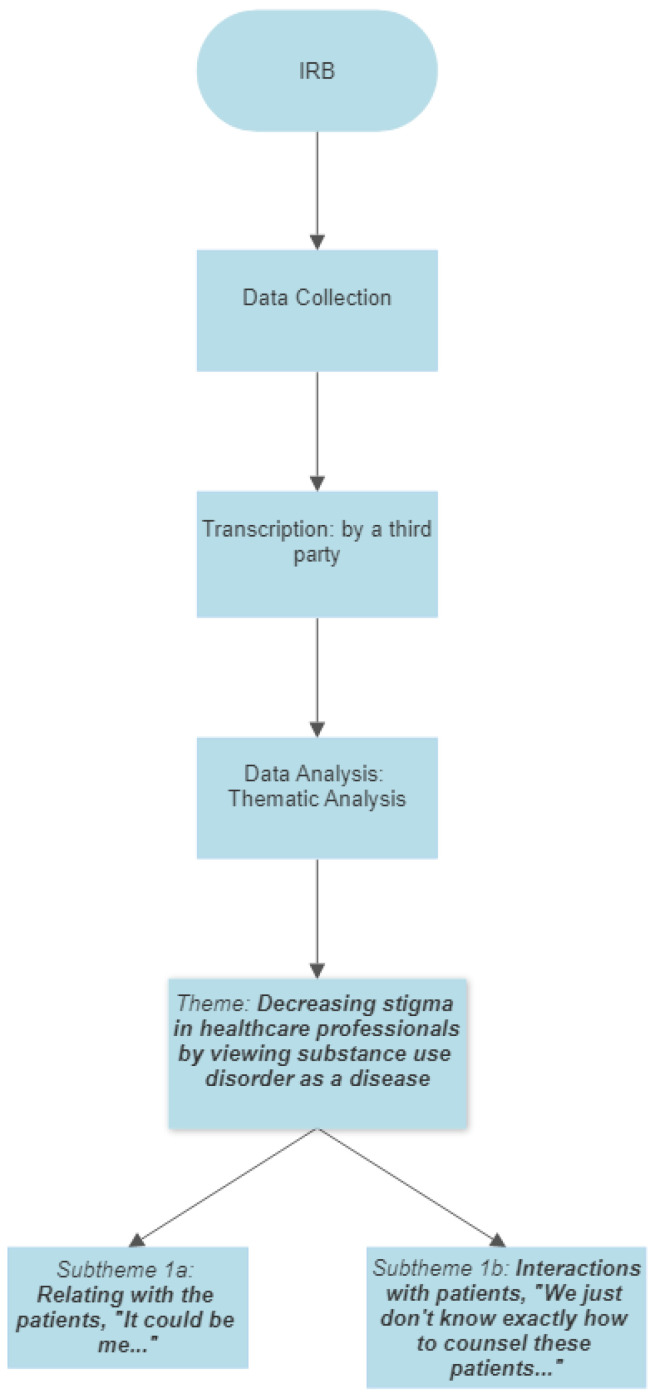
Flow chart for data the analysis of the data collected.

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
