# Peer review of "Healthcare Professional Students’ Perspectives on Substance Use Disorders and Stigma: A Qualitative Study"

_ijerph, 2022, doi:10.3390/ijerph19052776_

Round 1
Reviewer 1 Report
An interesting manuscript with good background and purpose. I think that the authors make the assumption that the readership will already understand qualitative research and the terms used including "thematic saturation" are already known to the reader. I think explaining the background of your methods for this qualitative research would improve this manuscript. I also think that if the authors included a flowchart of their methods it would be beneficial.
I don't completely agree that the conclusion of the paper is that pharmacy students need training in counseling patients with SUD. I think the focus group comments throughout the manuscript convey that these students have at least been trained in how to be empathetic and that a person with SUD may be reluctant to be honest for fear of being judged. I think the authors should take time to describe the current curriculums in the medicine and pharmacy programs and explain the current didactic and/or experiential training these students have already had in their programs and how this training has already prepared them for interacting with patients with SUD.
I think the authors should provide more of an explanation of how the curriculum for medical students differs from pharmacy students by comparing and contrasting these programs. This may help to explain why the authors believe that the pharmacy students are less equipped to deal with patients with SUD.
The authors state the following, "Although the participants were from a homogenous geographic location, ethnicity, and gender identity, this sample enabled this study to examine the participants' perceptions of the topic in some depth. It would be valuable to conduct further work inclusive of participants with varying demographics, backgrounds, and geographic areas of the US." This may be true but I still think it would be helpful to explain the overall demographics of these programs and not just the students in the focus groups.
Who conducted the transcriptions? Was it someone within the schools or outside of the schools?
For the following statement, "The study reveals that stigma toward people with SUDs is a barrier to accessing treatment and receiving adequate care. Healthcare professionals who do not understand the nature of SUD and intentionally combat their own stigma are likely to provide sub-optimal care and perpetuate stigma. They also may be more likely to engage in risky substance use themselves." I don't see any evidence or support for this statement based on this study. I also don't recall this being supported in the background. The authors should provide support for this conclusion. The comments reported from the focus group convey to me that these students have some idea how to interact with patients with SUD and understand the stigma rather than convey this stigma in their attitudes/comments.
What intentional instruction of how to interact with SUD patients is already in the medical and pharmacy curriculums? What other general skills that would help with dealing with SUD patients (e.g. empathy, counseling skills) have been taught in the curriculum that could help with SUD students? My impression is that these students may need more training but they seem to already have some semblance of the needs of SUD patients. The authors should spend more time in the manuscript describing the pharmacy and medicine curriculums and reflect upon what the students are already learning that could be improved upon to apply specifically to SUD.
Author Response
1)interesting manuscript with good background and purpose. I think that the authors make the assumption that the readership will already understand qualitative research and the terms used including "thematic saturation" are already known to the reader. I think explaining the background of your methods for this qualitative research would improve this manuscript. I also think that if the authors included a flowchart of their methods it would be beneficial.
Response: Thank you for your valuable suggestion. We amended the text by described the “thematic saturation”. We also included a flowchart that explains the methods. Thank you again for this recommendation that strengthen our manuscript. We added this paragraph “Contrary to the quantitative data where power or a statistical formula is used for determining the sample size, qualitative research uses "theoretical saturation" or "redundancy." In this theoretical saturation, the team continues recruitment of the participants and conducts FG until no new insights on the topic are presented in the FG. Thus, the team has exhausted the range of views on the specific topic. Therefore, conducting additional FG after reaching theoretical saturation will not result in new information on the topic.”
I don't completely agree that the conclusion of the paper is that pharmacy students need training in counseling patients with SUD. I think the focus group comments throughout the manuscript convey that these students have at least been trained in how to be empathetic and that a person with SUD may be reluctant to be honest for fear of being judged. I think the authors should take time to describe the current curriculums in the medicine and pharmacy programs and explain the current didactic and/or experiential training these students have already had in their programs and how this training has already prepared them for interacting with patients with SUD.
Response: Thank you for this insightful recommendation. We added a paragraph that describe the Pharm.D. and M.D. curricula at our institution.
We amended the manuscript: “The focus in recent years has shifted to include training on SUDs in the curriculum due to their prevalence. However, it is not clear to what extent, when in the curriculum, and how this training is being assessed. The Doctor of Pharmacy (Pharm.D.) and Doctor of Medicine curriculum at the University of Tennessee Health Science Center is comprised of a four year curriculum, including both didactic classes and experiential education. During the Pharm.D. curriculum, the students can enroll in elective courses that explore various disease states or provide additional interprofessional activities. Student pharmacists study pharmacotherapy, medicinal chemistry, and pharmacology of opioids throughout the Pharm.D. curriculum. Education about practices to engage effectively with patients is limited, and if received, is usually incorporated into the therapeutic modules throughout the curriculum.[8] It is not clear how well the training Pharm D. and M.D. students receive prepares them for working with patients with SUDs.”
I think the authors should provide more of an explanation of how the curriculum for medical students differs from pharmacy students by comparing and contrasting these programs. This may help to explain why the authors believe that the pharmacy students are less equipped to deal with patients with SUD.
Response: We value your suggestions. We made the changes in the methods and described the Pharm.D. curriculum. “The focus in recent years has shifted to include training on SUDs in the curriculum due to their prevalence. However, it is not clear to what extent, when in the curriculum, and how this training is being assessed. The Doctor of Pharmacy (Pharm.D.) and Doctor of Medicine curriculum at the University of Tennessee Health Science Center is comprised of a four year curriculum, including both didactic classes and experiential education. During the Pharm.D. curriculum, the students can enroll in elective courses that explore various disease states or provide additional interprofessional activities. Student pharmacists study pharmacotherapy, medicinal chemistry, and pharmacology of opioids throughout the Pharm.D. curriculum. Education about practices to engage effectively with patients is limited, and if received, is usually incorporated into the therapeutic modules throughout the curriculum.[8] It is not clear how well the training Pharm D. and M.D. students receive prepares them for working with patients with SUDs.”
The authors state the following, "Although the participants were from a homogenous geographic location, ethnicity, and gender identity, this sample enabled this study to examine the participants' perceptions of the topic in some depth. It would be valuable to conduct further work inclusive of participants with varying demographics, backgrounds, and geographic areas of the US." This may be true but I still think it would be helpful to explain the overall demographics of these programs and not just the students in the focus groups.
Response: Thank you for this insightful recommendation. We described the state of Tennessee from the demographics point of view. Furthermore, we added one sentence regarding the College of Pharmacy that has three campuses: Memphis, Nashville, and Knoxville. This recommendation strengthened our study.
“The social and economic diversity within Tennessee communities presents unique challenges related to the opioid epidemic and public health practice. Tennessee represents a state with diverse geography, spanning from Memphis (large minority population) to Nashville (urban and diverse), to Knoxville (a much smaller city, to numerous rural Appalachian areas). Unfortunately, Tennessee continues to be one of the worst states impacted by the opioid epidemic, with the number of heroin overdose-related deaths increasing over 300% in the last three years.”
“The College of Pharmacy has three campuses: Memphis, Nashville, and Knoxville, while the College of Medicine has one campus located in Memphis.”
Who conducted the transcriptions? Was it someone within the schools or outside of the schools?
Response: Thank you for this clarification. The original manuscript emphasized that the transcription was done by a professional company to minimize any bias. “All the participants agreed to be audio recorded and verbatim transcription was conducted by an objective professional company to minimize bias.”
For the following statement, "The study reveals that stigma toward people with SUDs is a barrier to accessing treatment and receiving adequate care. Healthcare professionals who do not understand the nature of SUD and intentionally combat their own stigma are likely to provide sub-optimal care and perpetuate stigma. They also may be more likely to engage in risky substance use themselves." I don't see any evidence or support for this statement based on this study. I also don't recall this being supported in the background. The authors should provide support for this conclusion. The comments reported from the focus group convey to me that these students have some idea how to interact with patients with SUD and understand the stigma rather than convey this stigma in their attitudes/comments.
Response: We are so grateful for your recommendation that helped us clarify certain aspects of the manuscript. The manuscript has been improved considerably because of your suggestion.
“The expertise from multiple providers is often necessary to provide the full scope of care a patient with SUDs needs. For example, it is not unlikely that a patient will require treatment for diabetes from a physician, a social worker who leverages support and resources, a counselor who treats substance misuse, a psychotherapist for co-occurring mental health disorders, and a pharmacist who provides medication. All healthcare providers are likely to treat patients who have active substance misuse and therefore must be trained to recognize symptoms and interact effectively with patients to adequately assess treatment needs and ensure that patients access appropriate care. Otherwise, patients are likely to receive suboptimal care and become more ill. For example, a retrospective study of patients hospitalized with infective endocarditis associated with injection drug use found high rates of readmission, recurrent disease, and death.[6] Less than a quarter (23.7%) of the patients had received addiction consultation, addiction was mentioned in only 55.9% of the discharge summary plans, and 7.8% of the patients had a plan to receive medication- assisted treatment.
What intentional instruction of how to interact with SUD patients is already in the medical and pharmacy curriculums? What other general skills that would help with dealing with SUD patients (e.g. empathy, counseling skills) have been taught in the curriculum that could help with SUD students? My impression is that these students may need more training but they seem to already have some semblance of the needs of SUD patients. The authors should spend more time in the manuscript describing the pharmacy and medicine curriculums and reflect upon what the students are already learning that could be improved upon to apply specifically to SUD.
Response: Thank you for these recommendations. We amended the text as recommended above.
Reviewer 2 Report
Dear authors,
in your study you are dealing with an important aspect of health care research. However, the manuscript still shows some gaps in the presentation of the study, its rationale and execution, especially the data analysis. I hope the following comments will help to further develop the manuscript.
Abstract: Your abstract still contains too much unimportant information about the methodology and the results are not presented concisely enough. Reading the abstract alone, it is obvious that the results are not surprising, as the two themes that emerged from the data analysis ("Thematic Analysis revealed two themes: (1) Healthcare professional students perceptions of Substance Use Disorders, and (2) Interactions with patients, "We just don't know exactly how to counsel these patients...") correspond to the research question (to characterize their collective perspectives about patients with SUDs and how that informs the care they provide). Try to name directly the results regarding the perspectives on patients with SUDs as well as the forms of interaction with these patients resulting from the perspectives.
Introduction: In your introduction, you have not described the importance of collaboration between prescribing practitioners and patients in enough detail. If I understand correctly, the starting point of your study is that there is stigmatization towards SUD patients and then the aim of the study is to display these stigmatization and derive implications for education of medical students (e.g. advise on communication with SUD patients etc.).
The connection between the collaboration between prescribing practitioner and pharmacist, which is so emphasized at the beginning of the introduction and at the end of the introduction, and the descriptions in between is unfortunately not clear at all. What is it about? About the collaboration between prescribing practitioner and pharmacist, or about stigmatization processes towards SUV patients on the part of the medical profession?
Methodology: "Stigma conceptualisation" should be explained and described in the introduction or in the state of the research. How did the concept inform the research question and the focus of the focus groups? In the methodology section, you should describe how the concept informed data analysis.
The examples of questions asked in the focus groups suggest that only "conscious" stigmatization was asked about. It can be assumed that stigmatization also occurs unconsciously. What does the literature say about this, and were you also able to capture such subliminal stigmatization through your method of analysis or only the directly expressed stigmatization?
Data Analysis: The descriptions in this section are too general. Can this also be related to your specific approach? How did the codes come about? Were further subcodes added inductively?
Results: "Nineteen participants were White, 14 were from the College of Medicine (COM), and 17 were from the College of Pharmacy (COP)." --> Why is it important whether the participants were white, and why is this mentioned in the same sentence as the colleges? If the skin colour of the participants in the focus groups is in any way important for the results of this study, then this should also be addressed in the theoretical part beforehand.
The results were substantiated with far too many direct quotes. The interpretations of the results are not profound and the results are not very abstract. The examples given among the topics also diverge widely in terms of content. It would be worthwhile to differentiate the results even further and present them more concisely. Especially regarding the questions. What are the stigmatizations? Are there any at all? How does this influence the interaction with the patients? Are there "common" strategies that are used or suggested by the medical students?
Discussion: "A novel finding of this study is the collective perspective of participants and that their own potential stigma was mitigated by personally relating to the circumstances of patients with a SUD." --> Did you even grasp the stigmatizations that existed before?
Overall, the study is very confusing and lacks depth. The interpretations do not go very far and the actual question is unclear and does not seem answered to me. The embedding in the theoretical discussions on stigmatization, patient compliance and the impact of interaction and communication between health care providers and patients is completely missing.
Author Response
Introduction: In your introduction, you have not described the importance of collaboration between prescribing practitioners and patients in enough detail. If I understand correctly, the starting point of your study is that there is stigmatization towards SUD patients and then the aim of the study is to display these stigmatization and derive implications for education of medical students (e.g. advise on communication with SUD patients etc.).
Response: Thank you for this valuable suggestion. In the view of the other reviewers’ recommendations, we made changes in the text.
“This manuscript focuses on one emergent theme and two subthemes align with the study objectives. The theme and sub-themes will be described in conjunction with participant quotations and excerpts from the FGs. The theme (1) Decreasing stigma in healthcare professionals by viewing substance use disorder as a disease , and two sub-theme: Subtheme 1a: Relating with the patients, “It could be me…” Subtheme 1b: Interactions with patients, “We just don’t know exactly how to counsel these patients…”
The connection between the collaboration between prescribing practitioner and pharmacist, which is so emphasized at the beginning of the introduction and at the end of the introduction, and the descriptions in between is unfortunately not clear at all. What is it about? About the collaboration between prescribing practitioner and pharmacist, or about stigmatization processes towards SUV patients on the part of the medical profession?
Response: We value your insightful suggestion. We made changes throughout the manuscript.
“The Current Study
Given the wide gap in accessing treatment and the role of stigma in perpetuating suboptimal care for patients with SUDs, there is a critical need for investigating health care students’ experiences with treating patients with SUDs, and prepare them to provide services that transcend stigma. The collaboration between prescribing practitioner and pharmacist may be a key to identifying barriers and implementing initiatives that improve access to and quality of care provided, especially during times of a global pandemic which has exacerbated an already existing epidemic. However, data are lacking about the collective voice of medical and pharmacy students’ perspectives and confidence managing patients with SUD. Thus, the objective of this study was to characterize future healthcare practitioner perspectives about what works best to reduce stigma and interact effectively with patients with SUDs when providing care. “
Methodology: "Stigma conceptualisation" should be explained and described in the introduction or in the state of the research. How did the concept inform the research question and the focus of the focus groups? In the methodology section, you should describe how the concept informed data analysis.
Response: We value this recommendation. The changes recommended strengthened our manuscript.
For Methods: To better gain insights into the concept of stigma, the first question of the FG was, "How would you define stigma in your own words?" This opening question allowed the participants to discuss their views. Furthermore, they also provided various situations in which they could portray stigmatizing behaviors observed throughout their clinical work.
The examples of questions asked in the focus groups suggest that only "conscious" stigmatization was asked about. It can be assumed that stigmatization also occurs unconsciously. What does the literature say about this, and were you also able to capture such subliminal stigmatization through your method of analysis or only the directly expressed stigmatization?
Response: Thank you for this question. Unfortunately, our data did not capture the subliminal stigmatization.
Data Analysis: The descriptions in this section are too general. Can this also be related to your specific approach? How did the codes come about? Were further subcodes added inductively?
Response: Thank you for this clarification. We followed the Thematic Analysis and the 6 steps described by Braun and Clarke. “Similar codes were grouped into categories.[5] All the categories were clustered and analyzed to uncover the major themes.”
Results: "Nineteen participants were White, 14 were from the College of Medicine (COM), and 17 were from the College of Pharmacy (COP)." --> Why is it important whether the participants were white, and why is this mentioned in the same sentence as the colleges? If the skin colour of the participants in the focus groups is in any way important for the results of this study, then this should also be addressed in the theoretical part beforehand.
Response: Thank you for this recommendation. We provided the demographics of these participants to show the heterogeneity of our participants. We amended the text.
The results were substantiated with far too many direct quotes. The interpretations of the results are not profound and the results are not very abstract. The examples given among the topics also diverge widely in terms of content. It would be worthwhile to differentiate the results even further and present them more concisely. Especially regarding the questions. What are the stigmatizations? Are there any at all? How does this influence the interaction with the patients? Are there "common" strategies that are used or suggested by the medical students?
Discussion: "A novel finding of this study is the collective perspective of participants and that their own potential stigma was mitigated by personally relating to the circumstances of patients with a SUD." --> Did you even grasp the stigmatizations that existed before?
Response: We value your suggestion. Although there is plethora of studies in the healthcare about stigma, there were no previous studies conducted with medical and student pharmacists.
Reviewer 3 Report
Method:
There might be a risk of selection bias because there was no clarification to address the stigma among different education levels of the participants (maybe state what year the participants) or how to ensure that the participants are familiar with the topic. Furthermore, the topic that the participants are familiar with may need to be clarified.. either about SUDs or about the stigma or both.
The "Stigma Conceptualization" as referred to Link & Phelan (2001) is an extensive write-up. It will be helpful for the readers if the context of "stigma" in the FG guide is briefly elaborated.
Is it necessary to first do some sort of assessment of stigma or self-stigma?
Result:
Is there perhaps any subtheme emerged regarding the category of stigmatization? Because based on the question "Describe a situation when you interacted with a patient who was diagnosed with substance use disorder", the participants' stigmatization themes were not reported.
Suggest presenting each theme under a separate sub-topic for a better understanding of the result flow.
The themes are not really aligned with the objective of this study in which "to characterize future healthcare practitioner perspectives about patients with SUDs and the impact stigma may have on providing care". Suggest considering to address what type of stigma will give an impact on each of the perspectives.
Author Response
There might be a risk of selection bias because there was no clarification to address the stigma among different education levels of the participants (maybe state what year the participants) or how to ensure that the participants are familiar with the topic. Furthermore, the topic that the participants are familiar with may need to be clarified.. either about SUDs or about the stigma or both.
Response: Thank you for this clarification. The email was sent to the entire UTHSC. However, the medical students were in the terminal year. Therefore, we amended the text to reflect this information: “All the COM participants were in the fourth year of their degree, while the COP participants represented second, third, and fourth-year.”
The "Stigma Conceptualization" as referred to Link & Phelan (2001) is an extensive write-up. It will be helpful for the readers if the context of "stigma" in the FG guide is briefly elaborated.
Response: We value your recommendation. The text was amended: “To better gain insights into the concept of stigma, the first question of the FG was, "How would you define stigma in your own words?" This opening question allowed the participants to discuss their views. Furthermore, they also provided various situations in which they could portray stigmatizing behaviors observed throughout their clinical work.”
Is it necessary to first do some sort of assessment of stigma or self-stigma?
Response: Thank you for this suggestion. It would be a recommendation for future studies.
Result:
Is there perhaps any subtheme emerged regarding the category of stigmatization? Because based on the question "Describe a situation when you interacted with a patient who was diagnosed with substance use disorder", the participants' stigmatization themes were not reported.
Suggest presenting each theme under a separate sub-topic for a better understanding of the result flow.
The themes are not really aligned with the objective of this study in which "to characterize future healthcare practitioner perspectives about patients with SUDs and the impact stigma may have on providing care". Suggest considering to address what type of stigma will give an impact on each of the perspectives.
Response: Thank you for these valuable suggestions. We made changes in the themes and also developed a flow diagram that strengthened our manuscript. Thank you again for these insights.
Methods: The FGs were semi-structured so the discussions among participants could emerge naturally while covering the intended scope at the same time. Participants were encouraged to talk about what their observations were about what worked best to reduce stigma and help patients with SUDs. The data in the transcriptions that were coded for being solution-oriented (rather than describing the problem) were analyzed in the current study.
Results: This manuscript focuses on one emergent theme and two subthemes align with the study objectives. The emergent theme and sub-themes will be described in conjunction with participant quotations and excerpts from the FGs.
Theme 1: Decreasing stigma in healthcare professionals by viewing substance use disorder as a disease
The overall theme explains how the healthcare professional participants recommended decreasing stigma when working with their SUD patients by understanding them as a person who is suffering with a disease. They tended to convey a perspective that by relating personally to the circumstances that patients with SUDs experience they were able to reconceptualize stereotypes as human struggles. In doing so they were able to intentionally not allow themselves to automatically ascribe negative judgement and attitudes that otherwise may have acted as a barrier to interactions with the patients.. Two subthemes further explain healthcare professional participants experiences with the reconceptualization of SUD patients.
Round 2
Reviewer 2 Report
Unfortunately, the authors did not address my comments on the "results" section. This section still needs revision.
The results were substantiated with far too many direct quotes. The interpretations of the results are not profound and the results are not very abstract. The examples given among the topics also diverge widely in terms of content. It would be worthwhile to differentiate the results even further and present them more concisely. Especially regarding the questions. What are the stigmatizations? Are there any at all? How does this influence the interaction with the patients? Are there "common" strategies that are used or suggested by the medical students?